# Targeting Molecular Dysregulation in Ulcerative Colitis: A Paired Cellular Perspective on CD4^+^, CD8^+^, and IL-6 Immunohistochemistry

**DOI:** 10.3390/ijms262411773

**Published:** 2025-12-05

**Authors:** Roxana Elena Mirica, Andrei Coman, Monica State, Cristiana Popp

**Affiliations:** 1Department of Social Insurance Medicine, Faculty of Medicine, Carol Davila University of Medicine and Pharmacy, 020021 Bucharest, Romania; 2National Institute of Medical Expertise and Recovery of Work Capacity, 050659 Bucharest, Romania; 3Department of Gastroenterology, Private Healthcare Network, Regina Maria, 011603 Bucharest, Romania; 4Department of Pathology, Colentina Clinical Hospital, 020125 Bucharest, Romania; andreiicoman08@gmail.com (A.C.); brigaela@yahoo.com (C.P.); 5Department of Gastroenterology, Colentina Clinical Hospital, Faculty of Medicine, Carol Davila University of Medicine and Pharmacy, 020125 Bucharest, Romania; monicastate4@gmail.com; 6Department of Pathology, Faculty of Medicine, Carol Davila University of Medicine and Pharmacy, 020021 Bucharest, Romania

**Keywords:** ulcerative colitis, histologic healing, CD4 T cells, CD8 T cells, IL-6, mucosal immunity

## Abstract

Histological healing is increasingly recognized as a sensitive marker of disease remission in ulcerative colitis (UC). However, the dynamics of mucosal T lymphocytes and proinflammatory cytokines during healing remain incompletely understood. In this paired, within-subject observational study (retrospective analysis of paired biopsies), colonic biopsy sets from 20 adult UC patients were analyzed during active inflammation and at a subsequent time point of histologic healing. Immunohistochemistry was performed for CD3, CD4, CD8, and IL-6. Lymphocyte densities were quantified in intraepithelial and lamina propria compartments, while IL-6 expression was scored semi-quantitatively. Histological activity was assessed using the Geboes score. Intraepithelial CD4^+^ T cells significantly decreased during histologic healing (mean 6.8 → 3.75 cells/100 epithelial cells, *p* < 0.05), whereas lamina propria CD4^+^ cells remained variably persistent, suggesting ongoing immune regulation. Intraepithelial CD8^+^ cells increased during remission, indicating a potential reparative or surveillance role. IL-6 expression markedly declined in epithelial and stromal compartments during healing, reflecting resolution of mucosal inflammation. Correlation analyses revealed enhanced coordination between CD4^+^ and CD8^+^ cells in the healing phase, consistent with immune homeostasis. Histologic healing in UC involves compartment-specific shifts in T lymphocyte populations and a marked reduction in IL-6 expression, reflecting coordinated immune regulation beyond clinical remission. These findings highlight the potential of combined cellular and cytokine biomarkers to monitor mucosal healing and guide immunomodulatory therapies.

## 1. Introduction

Inflammatory bowel diseases (IBDs), including ulcerative colitis (UC) and Crohn’s disease, represent a global problem with a significant impact on the quality of life of affected patients, characterized by continuous mucosal inflammation of the colon [1]. It is expected that by 2030 the incidence and prevalence of IBD will significantly increase [2]. Traditionally, clinical and endoscopic remission have been the primary therapeutic goals in UC management. However, accumulating evidence suggests that histological healing represents a deeper and more durable state of remission, associated with a lower risk of relapse, hospitalization, colorectal neoplasia, and colectomy [3]. Consequently, histologic healing has emerged as an important endpoint in both clinical trials and daily practice.

Despite its growing relevance, the immunological mechanisms underpinning histologic healing remain incompletely understood. The intestinal mucosa is a highly dynamic immune environment, in which T lymphocytes play pivotal roles in orchestrating inflammation and repair. Among them, CD4^+^ helper T cells and CD8^+^ cytotoxic T cells contribute to both tissue injury and immune regulation [4]. In the active phase of UC, CD4^+^ T cells promote cytokine release and the recruitment of inflammatory cells [5], whereas CD8^+^ T cells may mediate epithelial cytotoxicity or, alternatively, support mucosal barrier restoration during resolution [6]. Understanding the spatial distribution and functional balance of these T-cell subsets across the epithelial and lamina propria compartments is essential for understanding immune regulation during mucosal healing [7].

Interleukin-6 (IL-6) is another critical player in IBD pathogenesis [8]. As a multifunctional proinflammatory cytokine [9], IL-6 promotes T-cell activation, B-cell differentiation, and epithelial proliferation [10,11]. Elevated IL-6 expression has been documented in active UC mucosa, correlating with disease severity and systemic inflammation [12]. Conversely, downregulation of IL-6 may reflect mucosal recovery and immune quiescence [13]. However, the compartmental expression of IL-6 across different cellular and structural contexts (epithelium, stroma and endothelium) during histologic healing has not been systematically characterized.

The mucosal immune landscape is further shaped by interactions among epithelial cells interconnected through tight junctions [14], Paneth-like cells, goblet cells and the cytokine milieu. Dysregulation at this level represents a central mechanism in the pathogenesis of IBD. Inflammation arises from genetic variants associated with the disease, such as *NOD2* and *ATG16L1* mutations that impair Paneth cell function [15,16], as well as from defects in tight junctions, which lead to increased intestinal permeability and bacterial translocation into the lamina propria [17] and the deficiency of MUC2 mucin secreted by goblet cells which change the composition of the mucus [18], resulting in the impairment of the mucosal protective barrier [19,20]. Epithelial restitution after inflammation involves a shift from immune activation toward resolution, marked by changes in the lymphocyte composition and cytokine signaling. Characterizing these transitions offers insights into the persistence of subclinical immune activity that may predispose one to relapse even after apparent histologic healing.

Therefore, the aim of this study was to evaluate the dynamics of CD4^+^ and CD8^+^ lymphocytes and IL-6 expression in colonic biopsies from UC patients during active inflammation and histologic healing. By analyzing matched biopsy sets from the same individuals at both disease stages, we sought to elucidate how mucosal immune cell distribution and cytokine expression evolve during healing, and to identify potential immunologic signatures of sustained remission.

### 1.1. Cellular and Molecular Mechanism

#### 1.1.1. Local Immunity: Macrophages, Mast Cells, and Eosinophils

The intestinal mucosal immune system maintains a delicate balance between tolerance and inflammation. Resident immune cells within the lamina propria continuously interact with the epithelium and microbiota, and disruption of these interactions contributes to the chronic inflammation characteristic of IBD [21]. Intestinal macrophages exhibit functional plasticity, oscillating between the proinflammatory M1 and the reparative M2 phenotypes [22]. In IBD, predominant M1 activation leads to increased secretion of IL-1β, TNF-α, and IL-6, amplifying inflammatory infiltration and fibrogenesis [23]. Mast cells, abundant in the intestinal mucosa, interact with the enteric nervous system and release histamine, tryptase, and TNF-α, thereby increasing epithelial permeability and modulating neuroimmune responses [24]. Eosinophils infiltrate the mucosa even during clinical remission, releasing cytotoxic granule proteins such as Major Basic Protein (MBP) and Eosinophil Cationic Protein (ECP), which can damage epithelial cells and sustain inflammation [25,26]. Altogether, macrophages, mast cells, and eosinophils not only reflect the inflammatory status but actively participate in the disease progression and chronicity.

#### 1.1.2. Cell–Cell Interactions and the Role of the Mesenchymal Compartment

Beyond epithelial and immune cells, the intestinal mesenchymal compartment plays a pivotal role in the pathogenesis of IBD through dynamic interactions with the inflammatory microenvironment [27]. Fibroblasts and myofibroblasts act as major effectors of tissue remodeling. Under the influence of proinflammatory mediators (TNF-α, TGF-β, IL-17), they proliferate and secrete extracellular matrix (ECM) components, promoting fibrosis and stricture formation [28,29]. Mesenchymal stromal cells normally support mucosal homeostasis but adopt a proinflammatory phenotype during chronic inflammation [30,31]. Single-cell RNA sequencing studies have identified fibroblast subsets involved in T-cell recruitment and resistance to anti-TNF therapy [32]. These fibroblast–immune–epithelial interactions contribute to pathological angiogenesis, ECM remodeling, and perpetuation of the inflammatory cycle. Thus, the mesenchymal compartment is an active participant rather than a passive bystander, representing an emerging therapeutic target in IBD.

#### 1.1.3. Cytokines and Molecular Signaling

Cytokines orchestrate immune responses and tissue remodeling in IBD. Dysregulation of proinflammatory and antifibrotic signaling pathways drives both chronic inflammation and fibrotic complications. The TGF-β/Smad pathway has a dual role: physiologically maintaining immune tolerance [33], and in inflammatory settings inducing fibroblast-to-myofibroblast transition and excessive ECM deposition. Overexpression of Smad7, which interferes with transforming growth factor-β1 (TGF-β1) signaling, is a characteristic feature of inflammatory bowel disease [34]. In addition, Smad7 inhibition has shown therapeutic potential in experimental models [35]. IL-17A, produced by Th17 cells, promotes collagen and heat shock protein 47 (HSP47) expression in fibroblasts [36]; however, clinical blockade of IL-17 has proven ineffective, highlighting its context-dependent functions [37]. TNF-α enhances TGF-β and tissue inhibitor of metallproteinase-1 (TIMP-1) expression, contributing to fibrosis [38]; anti-TNF agents reduce inflammation but fail to prevent strictures [39]. Th2 cytokines (IL-4, IL-13) which are overexpressed in fibrotic IBD, also drive collagen synthesis [40], whereas Th1 cytokines, particularly IFN-γ, exert antifibrotic effects by inhibiting TGF-β signaling [41,42,43]. Collectively, cytokine signaling in IBD represents a fragile balance between inflammatory and fibrogenic pathways, differing between Th1/Th17-dominant Crohn’s disease and Th2-skewed ulcerative colitis.

#### 1.1.4. Immunologic Receptors and Cellular Responses

Genetic susceptibility to IBD is closely linked to alterations in innate immune receptors that detect microbial components. NOD2 is a cytosolic receptor for muramyl dipeptide (MDP) [44]. NOD2 mutations impair Paneth cell defensin production [45] and autophagy [46], leading to defective microbial clearance. TLR5, a Toll-like receptor recognizing bacterial flagellin, maintains microbial homeostasis and epithelial defense; its reduced expression in severe ulcerative colitis and spontaneous colitis in TLR5-deficient mice highlight its protective role [47]. Osteopontin (Opn), a matrix glycoprotein with immunomodulatory activity, is elevated in both the serum [48] and intestinal tissue of IBD patients [49], correlating with disease severity and chronic inflammation [50]. Dysregulation of these innate immune receptors not only initiates aberrant inflammatory responses but also perpetuates mucosal injury, identifying them as promising targets for next-generation immunotherapies.

#### 1.1.5. Microbiota and Receptor-Mediated Signaling

The intestinal microbiota is a key determinant of immune homeostasis, and dysbiosis is a hallmark of IBD. Alterations in microbial composition influence the local metabolism, including bile acid profiles, thereby affecting immune signaling and epithelial integrity. Bile acid receptors such as G-protein-coupled bile acid receptor 1 (GPBAR1, also known asTGR5) [51] and the nuclear receptor FXR modulates cytokine secretion and barrier function [52]; dysbiosis reduces the natural ligands for these receptors, promoting inflammation and permeability. The transcription factor (RAR-related orphan receptor gamma) RORγt, crucial for Th17 differentiation, is indirectly regulated by microbial metabolites, emphasizing the microbiota–immune axis [53,54]. Beyond compositional shifts, dysbiosis also involves profound changes in the intestinal metabolome. Overall, the microbiome and its metabolites function as an “immunologic organ” shaping the balance between tolerance and inflammation, with distinct microbial and metabolic profiles characterizing Crohn’s disease and ulcerative colitis.

Although immune dysregulation in inflammatory bowel disease provides a general framework, the present study focuses specifically on ulcerative colitis. Therefore, we centered our analysis on UC-relevant markers—CD4^+^, CD8^+^, and IL-6—evaluated in paired biopsies collected during active inflammation and histologic healing.

## 2. Results

### 2.1. Histological Disease Activity

All 20 patients included in the study had paired colonic biopsies, one from an active disease phase and one from a subsequent histologic healing phase, using the Geboes score, which assesses the histological activity in ulcerative colitis. This grading system includes the following features: crypt architecture, chronic inflammation and eosinophilis in the lamina propria, intraepithelial neutrophils and lamina propria neutrophils, crypt destruction and surface epithelial injury [55]. The Geboes score ranges from 0 to 5.4. A simplified version indicates an active histological inflammation when the score is ≥3.1 and inactive disease of ulcerative colitis with a score less than 3.1 [56,57].

During the active phase, the Geboes score ranged from 1.2 to 5.4 (median ≈ 4.0), reflecting variable histologic inflammation. In contrast, during histologic healing, scores ranged from 0.3 to 2.0 (median ≈ 1.5), indicating significant mucosal improvement.

While in the active biopsies, the Geboes score ranged widely, the biopsies harvested during histologic healing had a narrow distribution, supporting the robustness of histologic remission as a defined endpoint, overall, the difference was statistically significant (*p* = 0.0000001) (Figure 1). The effect size was d ≈ 1.98, indicating a very strong effect.

Despite the overall group consistency, one patient had a notably low Geboes score of 0.3, in the histologic healing phase, potentially indicating a deeper state of mucosal quiescence. Conversely, three patients in the active group had Geboes scores ≤ 1.3, suggesting mild or patchy inflammation. Geboes scores showed a significant reduction between active inflammation and histologic healing (Wilcoxon signed-rank *p* < 0.05)

### 2.2. Distribution of T Lymphocyte Subsets

#### 2.2.1. Cd4^+^ Lymphocytes

These findings indicate that intraepithelial CD4^+^ cells correlate with active inflammation, while lamina propria CD4^+^ cells may persist beyond visible healing, possibly reflecting a regulatory or memory phenotype. Intraepithelial CD4^+^ cell counts had a median value of 5 during the active phase and 4 during histologic healing; this difference was not statistically significant (*p* = 0.08). However, we observed a larger standard deviation for the active phase (6.9 vs. 3.2 in the histologic healing phase), reflecting greater variability among patients. In contrast, lamina propria CD4^+^ cell counts did not consistently decrease during histologic healing: the mean remained similar (30.25 vs. 29.25), while the median increased significantly (10 to 30), suggesting a possible repopulating or regulatory behavior of CD4^+^ cells in healing tissue (*p* < 0.05). Intraepithelial CD4^+^ T-cell counts significantly decreased from the active inflammatory phase to the histologic healing phase, as demonstrated by the Wilcoxon signed-rank test (*p* < 0.05).

#### 2.2.2. CD8^+^ Lymphocytes

CD8^+^ T cells were consistently observed in both the intraepithelial and lamina propria compartments. During the active phase of ulcerative colitis, CD8^+^ cells were present in higher numbers compared to intraepithelial CD4^+^ cells (mean 11.25 vs. 6.8 cells/100 epithelial cells), reflecting their prominent role in mucosal immunity. Intraepithelial CD8^+^ cells increased significantly during histologic healing (mean 22.5 cells/100 epithelial cells, *p* < 0.05, Table 1, Figure 2 and Figure 3), suggesting a potential role in epithelial repair and immune surveillance once active inflammation subsides. Lamina propria CD8^+^ cell counts remained relatively stable (data in Table 1), indicating persistent immune presence in the stromal compartment during remission. This pattern highlights the dynamic role of CD8^+^ lymphocytes in both cytotoxic responses and tissue homeostasis during the transition from active inflammation to healing. Intraepithelial CD8^+^ T-cell counts significantly increased during histologic healing compared to the active inflammatory phase (Wilcoxon signed-rank *p* < 0.05).

Lamina propria CD8^+^ T cells were consistently observed across patients, showing a pattern similar to intraepithelial CD8^+^ cells. While exact counts were not quantified, their presence supports a role in mucosal immune surveillance during both active inflammation and histologic healing. Intraepithelial CD8^+^ T-cell counts showed a modest increase from the active phase to histologic healing. Although individual paired values varied across patients, the overall direction of change indicated higher CD8^+^ density during healing, consistent with the trend observed in the descriptive statistics.

### 2.3. Il-6 Expression

IL-6 immunoreactivity was detected in epithelial, stromal, and endothelial compartments of colonic biopsies during both the active inflammation and histologic healing phases. In the active phase, IL-6 staining was generally higher across all compartments, with median scores of 1.0–2.0 observed in the epithelium and stroma, and lower expression in the endothelium. The epithelial and stromal compartments frequently showed moderate–strong IL-6 positivity, reflecting active cytokine production during inflammation.

During histologic healing, IL-6 expression significantly decreased in the epithelial and stromal compartments (*p* < 0.01), whereas endothelial IL-6 staining remained low and changes were minimal. The majority of epithelial and stromal samples exhibited low or absent IL-6 staining (median scores close to 0–1), indicating the resolution of active inflammatory signaling. Endothelial IL-6 staining remained low in both phases, with minimal changes.

Overall, these findings demonstrate a marked reduction in IL-6 expression in the mucosa during healing compared to active disease, consistent with decreased inflammatory activity and cytokine production as the tissue recovered. IL-6 expression scores were significantly lower in the healing phase than in the active phase (Wilcoxon signed-rank *p* < 0.05).

### 2.4. Correlation Analysis

We used correlation analysis to explore the relationships between the CD4^+^, CD8^+^, and IL-6 expression patterns (Figure 4).

In the active phase, correlations between intraepithelial CD4^+^ and CD8^+^ counts were weak (Spearman’s ρ ≈ 0.21, *p* = 0.35), suggesting independent recruitment during inflammation.During histologic healing, the correlation strengthened significantly (ρ ≈ 0.62, *p* < 0.01), reflecting a more coordinated immune cell presence consistent with restoration of mucosal homeostasis.IL-6 expression correlated positively with intraepithelial CD4^+^ density (ρ = 0.55, *p* < 0.05) during active disease but lost significance in remission, highlighting its role in inflammation-driven immune activation.

These histologic and immunologic findings provide insight into the persistence and modulation of mucosal immunity, forming the basis for the discussion below on the immune dynamics and potential therapeutic implications.

Using the available aggregated paired dataset, a positive directional association was observed between CD4^+^ and CD8^+^ densities, while IL-6 expression showed an inverse directional relationship with lymphocyte counts. Because raw paired values were unavailable, exact rho coefficients, confidence intervals, and phase-specific within-subject correlations could not be calculated. As such, results are presented descriptively.

## 3. Discussion

### 3.1. Histologic and Immunologic Correlates of Mucosal Healing

Histologic activity has been increasingly recognized as a sensitive indicator of residual disease, and growing evidence supports histologic remission being associated with a lower risk of relapse, hospitalization, and colectomy. Our findings are consistent with previous studies suggesting that histologic healing should be considered a key therapeutic target in UC management.

We also observed minor overlaps at the lower end of the inflammatory scale. A small subset of patients with active disease had Geboes scores below 2.0, and one patient in the histologic healing phase had a score of 0.3, reflecting deep remission. These cases emphasize the heterogeneity of inflammation in UC, the potential for sampling variability, and the limits of clinical-endoscopic correlation. Histologic assessment adds an objective layer to disease evaluation and may be particularly valuable in therapeutic decision-making, especially when symptoms and endoscopic appearance do not align.

CD4^+^ T cells are helper T cells involved in immune activation. In the epithelium, they signal active inflammation and immune surveillance. In our group, intraepithelial CD4^+^ lymphocytes significantly decreased during histologic healing compared to the active phase (mean 6.8 → 3.75 cells/HPF; *p* < 0.05). The decrease was confirmed with a Wilcoxon signed-rank test, supporting the role of intraepithelial CD4^+^ cells as markers of active mucosal inflammation.

In contrast, lamina propria CD4^+^ cell density did not consistently decrease during healing. During the active phase, estimated counts ranged widely (mean 30.25 cells/HPF; median 10), reflecting heterogeneous immune infiltration. During histologic healing, the mean remained comparable (29.25 cells/HPF), but the median increased to 30 cells/HPF, suggesting persistence or repopulation of CD4^+^ lymphocytes. These findings indicate that intraepithelial CD4^+^ cells are closely tied to active inflammation, while lamina propria CD4^+^ cells may persist during healing, potentially reflecting immune regulation or memory.

### 3.2. CD8^+^ Lymphocyte Dynamics

Analysis of CD8^+^ lymphocytes revealed distinct patterns during active inflammation and histologic healing. Intraepithelial CD8^+^ cells increased significantly during healing (mean 11.25 → 22.5 cells/HPF; *p* < 0.05), suggesting a potential role in epithelial repair and immune surveillance. Lamina propria CD8^+^ cells remained relatively stable, indicating ongoing immune presence in the stromal compartment even after visible remission.

Correlation analysis showed that intraepithelial CD4^+^ and CD8^+^ cells were weakly correlated during the active phase (ρ ≈ 0.21, *p* = 0.35) but more strongly correlated during histologic healing (ρ ≈ 0.62, *p* < 0.01), reflecting coordinated immune interactions in remission. Lamina propria correlations were modest and variable, suggesting compartment-specific immune dynamics. Overall, CD4^+^ and CD8^+^ T cells play complementary roles depending on the mucosal compartment and disease stage.

### 3.3. IL-6 Expression and Cytokine Modulation

IL-6 immunoreactivity was highest in epithelial and stromal compartments during active inflammation (median 1–2) and decreased significantly during histologic healing (median 0–1; *p* < 0.01). Endothelial IL-6 expression was low in both phases. These findings illustrate a transition from active inflammatory signaling to immune regulation and homeostasis. The reduction in IL-6 parallels changes in T-cell populations, highlighting its role as a biomarker of mucosal inflammation and potential therapeutic target in UC.

### 3.4. Clinical Implications

Our data support incorporating histologic evaluation alongside endoscopic assessment to achieve deeper remission in UC. The persistence of certain immune cell subsets during histologic healing may explain the occurrence of relapse despite endoscopic quiescence. Targeted immunomodulatory strategies that reinforce regulatory CD4^+^ and reparative CD8^+^ functions while suppressing IL-6–driven inflammation may help sustain remission. The Janus kinase (JAK) which activates the signal transducer and activator of transcription (STAT) pathway is important to the pathogenesis of IBD [58]. An oral pan-JAK inhibitor (Tofacitinib) has shown clinical efficacy and is approved for the treatment of ulcerative colitis [59,60]. In the last few years, researchers have developed selective JAK 1 inhibitors such as Filgotinib and Upadatinib which showed improvements in moderate to severe UC [61,62].

### 3.5. Study Limitations

This study has several limitations. First, the sample size was relatively small (n = 20), which may limit the statistical power and generalizability of the findings. Second, the paired, within-subject comparative design precludes causal inference regarding the dynamic changes in mucosal T-cell populations and IL-6 expression over time. Third, only a limited set of immune markers (CD4, CD8, and IL-6) was analyzed; inclusion of additional cytokines and functional markers (e.g.,TNF-α, TGF-β, FOXP3, IL-17, granzyme B) could provide a more comprehensive understanding of immune regulation in ulcerative colitis. Fourth, inter-patient heterogeneity, despite paired biopsies, may influence immune cell distributions and histologic responses. Finally, the semi-quantitative nature of immunohistochemistry may not capture the full complexity of the cellular and cytokine dynamics, which could be further elucidated using multiplexed immunostaining or spatial transcriptomics.

Despite these limitations, our findings contribute to a growing understanding of immune modulation in UC histologic healing. Future research should integrate immunohistochemistry, transcriptomic, and spatial profiling to map mucosal immune niches in remission. Identifying immune signatures of durable healing could inform personalized therapeutic strategies and novel biomarkers for clinical monitoring.

In addition to the limitations already acknowledged (small sample size, limited number of markers, and the semi-quantitative nature of immunohistochemistry), several additional constraints must be considered.

First, concomitant medical therapies, which were not fully available for all patients, may confound immune cell densities and cytokine expression, and therefore the observed differences cannot be interpreted independently of potential treatment effects.

Second, sampling variability across colonic segments and over time may introduce heterogeneity in mucosal immune profiles, as paired biopsies—although obtained from comparable locations—cannot fully eliminate spatial or temporal variation.

Third, IL-6 immunohistochemistry provides limited cellular source specificity, and staining intensity does not allow robust discrimination between epithelial, stromal, and infiltrating immune cell contributions.

Given these factors and the retrospective design, the findings should be interpreted as associative, without implying mechanistic causality.

Another limitation is the lack of paired systemic inflammatory markers (such as serum IL-6 or CRP), which were not consistently available at both biopsy timepoints. This prevented direct comparison between mucosal immune changes and systemic inflammation. Future prospective studies integrating matched histologic, immunohistochemical, and laboratory parameters will be essential to clarify these mucosal–systemic relationships.

### 3.6. Future Perspectives and Research Directions

Inflammatory bowel disease (IBD) remains a complex disorder involving dysregulated immune responses, epithelial barrier dysfunction, mesenchymal activation, and microbiota alterations. Building on the findings of this study, future research should prioritize the following directions:Targeted therapeutics: Development and refinement of inhibitors for Smad7, MEK [63], and specific cytokines, with careful evaluation of safety and efficacy in defined patient subpopulations.Microbiota modulation: Exploration of bile acid receptor agonists and microbiome-targeted therapies to restore immune homeostasis and reduce mucosal inflammation [53].Single-cell and spatial omics: Mapping novel cellular subtypes and their interactions to identify predictive biomarkers of disease progression and therapeutic response [64].Integrative approaches: Consideration of social determinants of health to guide personalized preventive and therapeutic strategies [65].Translational models: Use of patient-derived organoids, co-culture systems, and humanized mouse models for mechanistic studies and preclinical drug testing [66].

This forward-looking framework emphasizes the integration of mechanistic insights with clinical and environmental data, supporting the development of precision-guided interventions and improved patient outcomes in IBD.

## 4. Materials and Methods

### 4.1. Study Design and Population

To investigate mucosal immune changes during active inflammation and subsequent recovery, we performed immunohistochemical analysis of T-cell subsets and IL-6 expression on paired colonic biopsies from the same patients. The study is a paired, within-subject observational analysis (retrospective review of biopsy pairs), designed to compare immune cell densities and cytokine expression between an active disease timepoint and a later histologic healing timepoint in each patient. Twenty adult patients with a confirmed diagnosis of UC of whom 10 were women and 10 were men, were recruited from the Gastroenterology Department of Colentina University Hospital, Bucharest and all their biopsies were evaluated, for diagnosis and research purposes in the Pathology Department of the same hospital. The mean age of the participants was 41 years (range 18–75). Patients had established disease at the time of inclusion, with a disease duration ranging from 6 months to 14 years. For each patient, two sets of colonic biopsies were analyzed: the first collected during active inflammation at the start of the study in 2011, and the second obtained during histologic healing, after a variable time interval determined by each patient’s clinical course (ranging from several months to several years). This paired sampling allowed a direct comparison of immune cell populations and cytokine expression between active and healed mucosa in the same individuals. The study protocol was approved by the Institutional Review Board, and written informed consent was obtained from all participants prior to inclusion.

### 4.2. Inclusion and Exclusion Criteria

Eligible participants were aged between 18 and 75 years and had a confirmed diagnosis of UC for at least six months, based on standard clinical, endoscopic, and histological criteria. Patients were required to have undergone a recent colonoscopy with available biopsy samples.

The exclusion criteria included: (i) concurrent diagnosis of Crohn’s disease or indeterminate colitis; (ii) history of colonic tuberculosis, colorectal cancer or dysplasia; (iii) recent use of investigational drugs; and (iv) presence of active systemic infection at the time of sampling.

### 4.3. Study Groups and Sampling Strategy

For each patient, two independent sets of biopsies samples were analyzed:Active phase samples, obtained during a documented clinical and endoscopic flare (Mayo endoscopic sub-score 2–3).Histologic healing samples, collected later from the same colonic segments once patients achieved endoscopic remission (Mayo sub-score 0) and were clinically asymptomatic.

Histologic healing was defined as a Geboes score < 2.0, corresponding to the absence of neutrophilic infiltration, crypt destruction, abscesses, erosions, or ulceration. Thus, each patient served as their own control, allowing direct comparison between inflammatory and remission states.

### 4.4. Clinical and Endoscopic Assessment

Disease activity, in both moments was assessed using the Mayo score, including both clinical parameters and endoscopic findings. Patients with active UC had a Mayo endoscopic sub-score of 2 or 3 and reported increased stool frequency, urgency, and rectal bleeding. In the histologic healing moment, they had a Mayo endoscopic sub-score of 0 and were asymptomatic at the time of biopsy.

### 4.5. Biopsy Collection and Processing

Colonic biopsies were obtained duringastandard colonoscopy from predefined locations (rectum, sigmoid, and descending colon). At least two mucosal samples per site were collected using cold forceps. The specimens were immediately fixed in 10% neutral buffered formalin and processed for histological evaluation.

### 4.6. Histopathological Evaluation

Formalin-fixed, paraffin-embedded (FFPE) tissue blocks were sectioned at 3 μm thickness and stained with hematoxylin and eosin (H&E) for initial histologic assessment. Two experienced gastrointestinal pathologists independently evaluated the samples (including the endoscopic scores of activity) while blinded to both the clinical data and the disease phase (active vs. histologic healing) using the Geboes score. Any discrepancies in Geboes scoring were resolved through joint review and consensus (Figure 5 and Figure 6).

For immunohistochemical (IHC) analysis, the same pathologists quantified T-cell subsets (CD3^+^, CD4^+^, CD8^+^) and IL-6 expression under the same blinding conditions, without knowledge of the sample’s disease phase. Discrepancies between observers in IHC counts were similarly resolved by joint review and consensus to ensure accuracy and reproducibility.

### 4.7. Immunohistochemical Analysis

In addition to H&E, immunohistochemical (IHC) staining was performed on each biopsy block using the following antibodies (Table 2):CD3 (pan–T-cell marker)—to evaluate the entire T lymph cell population;CD4 (helper T-cell marker);CD8 (cytotoxic T-cell marker);IL-6 (proinflammatory cytokine).

IHC staining was carried out on an automated platform (Leica Bond), according to the manufacturer’s protocols. After deparaffinization and rehydration, antigen retrieval was performed using citrate buffer (pH 6.0) or EDTA (pH 9.0), depending on the antibody. Visualization was achieved using DAB (3,3′-diaminobenzidine) as chromogen, and counterstaining was performed with hematoxylin.

The density of T lymph cells (CD3^+^, CD4^+^, CD8^+^) was scored in the hot spot in the lamina propria (number of positive cells on one high-power field with a diameter of 0.55 mm) and in the intraepithelial compartment (number of inflammatory cells reported at 100 epithelial cells). The hot-spot was identified by both pathologists in a bi-tete examination and included the area that visually had the densest infiltrate with T lymph-cells (CD3 positive). The value represents the average of the adjacent fields that included the area o lamina propria considered hot spot. For the count of the cells, Bland–Altman analysis showed minimal bias (mean difference = 1.2 cells) with narrow limits of agreement (−3.5 to 5.9 cells), indicating very high inter-observer agreement. All disagreements were solutioned by bi-tete examination.

CD3 antibodies were included as a pan–T-cell marker to confirm the overall presence and distribution of T lymphocytes in the examined mucosal compartments. CD3 staining was used qualitatively for verification purposes only, and no quantitative analyses were planned for this marker. Accordingly, CD3 data are not reported, and all statistical analyses focus exclusively on CD4^+^ and CD8^+^ subsets, which directly align with the study objectives; however, quantitative analyses focused specifically on CD4^+^ and CD8^+^ subsets, which are directly relevant to the study objectives.

IL-6 expression was assessed semi-quantitatively, in epithelial and endothelial cells, and in the inflammatory infiltrates from the stromal compartment. The results were expressed as categorical scores (0—absent; 1—mild; 2—moderate; 3—strong) staining intensity.

### 4.8. Ethical Considerations

The study protocol was approved by the Institutional Review Board of Colentina University Hospital. All patients provided written informed consent prior to inclusion. All procedures were conducted in accordance with the Declaration of Helsinki and relevant institutional guidelines.

### 4.9. Statistical Analysis

Descriptive statistics were used to summarize the demographic and clinical characteristics. Differences in histological parameters between groups were assessed using non-parametric tests (Mann–Whitney U test for continuous variables; Fisher’s exact test for categorical variables). Relationships between CD4^+^, CD8^+^, and IL-6 expressions were assessed using Spearman’s rank correlation. A *p*-value < 0.05 was considered statistically significant. The immunohistochemical characterization of CD4^+^, CD8^+^, and IL-6 expression allowed comparative analysis of mucosal immune activity during active inflammation and histologic healing in ulcerative colitis. All analyses were performed on paired biopsy samples taken from the same patients during active inflammation and subsequent histologic healing. Therefore, Wilcoxon signed-rank tests were used for all paired comparisons, including CD4^+^ T-cell counts, CD8^+^ T-cell counts, IL-6 expression scores, and Geboes histologic scores. Categorical variables were compared using Fisher’s exact test. Correlations between CD4^+^, CD8^+^ and IL-6 expression were evaluated using Spearman’s rank correlation. A *p*-value < 0.05 was considered statistically significant.

Given the retrospective design and the lack of complete raw paired distributions for all endpoints, paired median changes with confidence intervals could not be reconstructed; therefore, results are presented as direction of change and corresponding *p*-values. The primary endpoints were the paired changes in intraepithelial CD4^+^ T-cell counts, CD8^+^ T-cell counts, and epithelial IL-6 expression between active inflammation and histologic healing.

Secondary endpoints included paired stromal changes for these markers.

Exploratory endpoints comprised the correlations between CD4^+^, CD8^+^, IL-6 expression, and histologic activity scores. The correlation analyses were prespecified as exploratory due to expected variability and limited sample size.

Spearman’s rank correlation coefficients were calculated using the available aggregated paired dataset. Because raw per-patient values were not accessible in the retrospective archive, it was not possible to compute correlation coefficients (rho), exact *p*-values, confidence intervals, or to perform phase-specific or within-subject correlation analyses. Therefore, correlations are reported descriptively based solely on their direction and statistical significance obtained from the preserved summary dataset.

## 5. Conclusions

This research demonstrates that histologic healing in ulcerative colitis (UC) is accompanied by significant modulation of the mucosal immune architecture. These findings indicate that mucosal healing represents not merely the absence of inflammation, but an active process of immune remodeling and balance restoration. The persistence of certain lymphocyte subsets even during remission suggests that mucosal immune regulation continues beyond endoscopic or clinical recovery. This emphasizes the need to integrate histologic and immunologic biomarkers into treatment monitoring and therapeutic endpoints. Integration of cellular and molecular data could ultimately lead to precision-guided immunomodulatory approaches in this disease. Future research should aim to characterize the phenotypic and functional plasticity of mucosal T cells using multiplex immunohistochemistry and spatial transcriptomics, to delineate proinflammatory versus reparative subpopulations.

## Figures and Tables

**Figure 1 ijms-26-11773-f001:**
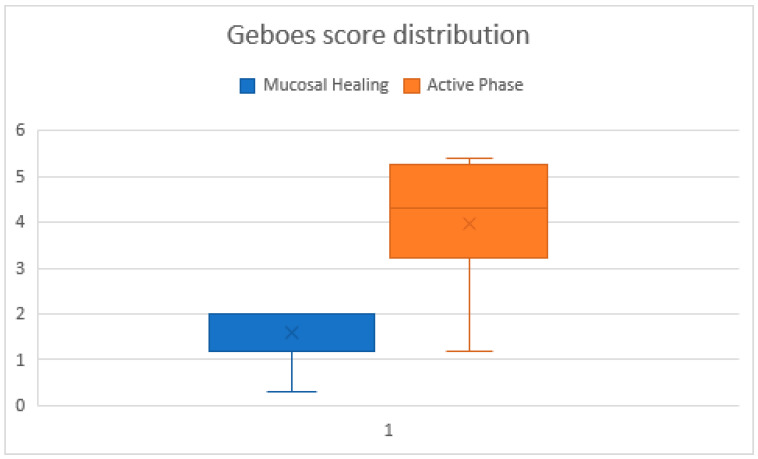
Comparative graphic of Geboes scores across biopsies in the healing and active phases. The chart illustrating individual Geboes scores for all biopsy samples. Lower and more stable scores correspond to the histologic healing phase, while higher variability and elevated scores indicate the active phase.

**Figure 2 ijms-26-11773-f002:**
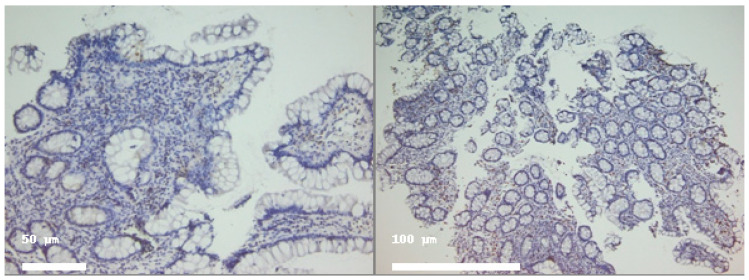
CD4 (**left**) and CD8 (**right**) immunostained in a patient with active disease. Note more frequent positive cells (brown) in both epithelial and stromal compartments. Magnification 100× (**left**) and 40× (**right**).

**Figure 3 ijms-26-11773-f003:**
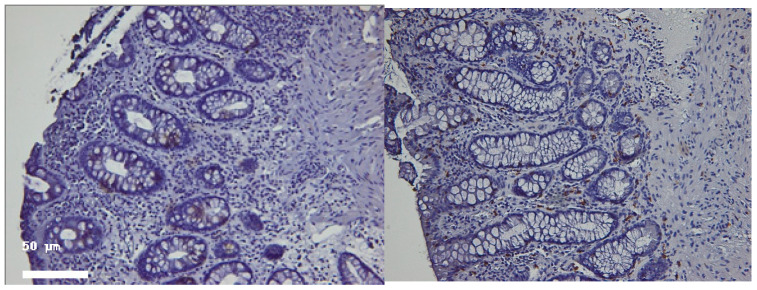
Representative colonic mucosal sections from patients with ulcerative colitis. CD4 (**left**) and CD8 (**right**) immunostained in the same patient in histologic healing phase. Note rare positive cells (brown) in both epithelial and stromal compartments. Hematoxylin–eosin staining of paired biopsies from the same patient. (**Left panel**): Active inflammation phase showing crypt architectural distortion, epithelial damage, and inflammatory cell infiltration. (**Right panel**): Histologic healing phase with restored crypt architecture and markedly reduced inflammatory infiltrate. Magnification 100×.

**Figure 4 ijms-26-11773-f004:**
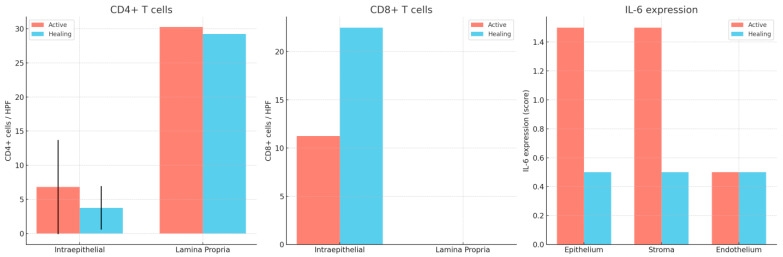
Correlative analysis between CD4^+^, CD8^+^, and IL-6 expression patterns in both active and healing phases.

**Figure 5 ijms-26-11773-f005:**
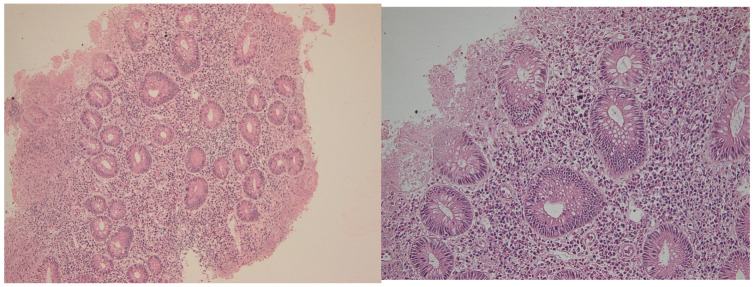
Active phase in a patient with ulcerative colitis. Note the severe inflammation, with erosion and significant regenerative epithelial changes. Hematoxylin–eosin 100× (**left**) and 400× (**right**).

**Figure 6 ijms-26-11773-f006:**
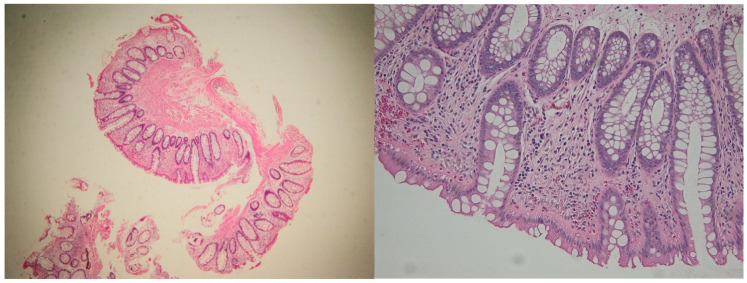
Samples from the same patient as above, but in histologic healing phase. Hematoxylin–eosin 100× (**left**) and 200× (**right**). Note the changes in cryptic architecture, but no inflammation, erosion or epithelial regenerative changes can be identified. Distribution of CD4^+^ and CD8^+^ T lymphocytes in active inflammation (**right**) and histologic healing (**left**) phases in representative patients. This figure illustrates cell counts per high-power field and is intended for comparison of abundance between disease stages; it does not depict correlations. Spearman’s rank correlation analyses are presented separately in Section 2.4.

**Table 1 ijms-26-11773-t001:** Quantitative summary of CD4^+^, CD8^+^ lymphocyte counts and IL-6 expression in active and histologic healing phases of ulcerative colitis.

Marker	Compartment	Active Disease (Mean ± SD)	Median (IQR)	Histologic Healing (Mean ± SD)	Median (IQR)
CD4^+^	Intraepithelial	6.8 ± 6.9	5 (3–9)	3.75 ± 3.2	4 (2–6)
	Lamina propria	30.25 ± —	10 (6–25)	29.25 ± —	30 (20–40)
CD8^+^	Intraepithelial	11.25 ± —	10 (7–15)	22.5 ± —	16 (12–25)
	Lamina propria	72.5±	71 (5–125)	82±	75 (5–175)
IL-6	Epithelium	1.5 ± —	1–2	0.5 ± —	0–1
	Stroma	1.5 ± —	1–2	0.5 ± —	0–1
	Endothelium	0.5 ± —	—	0.5 ± —	No significant change between phases

Note: Data are presented as mean ± standard deviation (SD) and median (interquartile range, IQR) where available. SD not available for some parameters due to limited sample variability or descriptive evaluation. Cell counts are expressed as number of positive cells per high-power field (HPF) in lamina propria and as number of positive cells/100 epithelial cells for the intraepithelial compartment. IL-6 staining intensity was semi-quantitatively scored (0 = absent, 1 = mild, 2 = moderate, 3 = strong). Together, these results suggest a shift from uncoordinated proinflammatory T-cell infiltrationtowardregulated immune surveillance and cytokine downregulation as healing progresses.

**Table 2 ijms-26-11773-t002:** Immunohistochemistry data sheet.

Antigen	Clone	Vendor	Catalog No.	Use	Retrieval Buffer/Pretreatment	Incubation/Dilution	Detection System
CD3	LN10	Leica Biosystems	PA0553	Ready-to-Use	BOND ER	BOND	BOND Polymer Refine Detection
CD4	4B12	Leica Biosystems	PA0427	Ready-to-Use	HIER	BOND	BOND
CD8	4B11	Leica Biosystems	PA0191	Ready-to-Use	HIER	BOND	BOND
IL-6	10C12	Leica Biosystems	NCL-L-IL6	Liquid concentrate	HIER	BOND	BOND

## Data Availability

The original contributions presented in this study are included in the article. Further inquiries can be directed to the corresponding author.

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
