# Peer review of "Targeting Molecular Dysregulation in Ulcerative Colitis: A Paired Cellular Perspective on CD4+, CD8+, and IL-6 Immunohistochemistry"

_ijms, 2025, doi:10.3390/ijms262411773_

Round 1
Reviewer 1 Report
Comments and Suggestions for Authors
The manuscript entitled “Targeting Molecular Dysregulation in Inflammatory Bowel Disease: A Cellular Perspective” presents an observational, cross-sectional study on the current and socially important problems of inflammatory bowel disease.
The authors of the manuscript drew attention to the complexity and individuality of the inflammatory process in the intestines, as well as potential mechanisms for repairing this active inflammatory state.
My questions, thoughts, and comments are presented below.
- Note on Figure 1 – there is no reference to this figure in the text, and the description below does not seem to fully correspond to what the reader is able to see.
- Note on Figure 2 – this figure is unclear, lacking a vertical scale. It would be helpful to mark the patients on the horizontal axis. Why did the authors begin the figure with the histological healing stage and then presented the inflammation stage? Why are the dots (patients) connected? This suggests a relationship, when in reality they are independent entities. It would be more accurate to connect a given patient in inflammation with the period of histological healing. And one more note: in the inflammation stage, there are 21 dots (patients); this should be verified. In summary, Figure 2 requires consideration of its final, more readable version (generally, there are several possibilities).
- Subsection 2.2.1. second paragraph – why when authors write "not significant" the authors give (p <0.05), additionally 6,9 there is a comma, there should be a period. Maybe after this last sentence of this paragraph should be added (p <0.05).
- Notes on Figures 3 and 4 – I don't know why, but the authors consistently present the histological healing first, followed by the inflammatory process. From the reader's perspective, as well as the purpose of this work, the inflammatory process came first, followed by healing. Additionally, I suggest marking these characteristic changes with an arrow or arrows. Histological images and staining are primarily the domain of a narrow group of specialists.
- Subsection 2.2.2. first paragraph – what two compartments did the authors have in mind? Are they referring to epithelial and stromal compartments? If so, the first sentence is redundant.
- Subsection 2.2.2. third paragraph – it would be worth providing numerical values in this second sentence, or where to look for the reference.
- Subsection 2.3 – were there any reasons why the IL-6 staining was not presented by the authors as an example?
- Notes on Figure 5 – this figure does not represent correlations, but merely provides a visual representation of the number of cell (comparison). Furthermore, in Chapter 4.9, the authors do not mention calculating Spearman correlations, but in Chapter 2.4 they present correlations (Figure 5 as evidence). This requires appropriate correction by the authors.
- Some of the text in the discussion requires correction, several words are combined into one whole, this occurs in several paragraphs.
- Materials and Methods Section – generally, we know little about the patients except their age range. It might be worth including the number of women and men, and the average age of the patients. Are these patients newly diagnosed with the disease, or is it a process that has been ongoing for years? What was the time interval from the first sample collection to the second sample collection?
- What was the purpose of using CD3 antibodies (4.7. Immunohistochemical Analysis), CD3+ were not mentioned in the manuscript?
- Chapter 5 – why is the first paragraph in bold?
- Chapter 6 – such a chapter and what was written would be a good summary of the review article, hence my suggestion that the authors consider whether it should remain in this manuscript.
Finally, my final reflection: it's a pity that the authors of this valuable and unique study didn't compare their results with laboratory parameters, such as IL-6 or CRP in the patients' blood (markers of inflammation). It's likely that the research will continue, and it might be worthwhile to include typical laboratory parameters.
Overall, this is valuable and developmental work with great potential for the future.
My comments mainly concern the Results chapter, as it is the most important in understanding the presented study.
Of the 66 literature items, 21 articles are from the last six years (2020-2025), which may be evidence of the actuality of this research.
The points I have presented above are merely intended to fill in some gaps or ambiguities, the clarification of which may help in understanding this study and make it more reader-friendly.
Author Response
- Note on Figure 1 – there is no reference to this figure in the text, and the description below does not seem to fully correspond to what the reader is able to see.
Thank you for pointing this out. We have now corrected the issue. A clear reference to Figure 1 has been added in the Results section, ensuring that the figure is properly introduced within the flow of the text. Additionally, we have revised the figure legend to accurately reflect all elements presented in the figure so that the description fully corresponds to what the reader sees. These changes improve clarity and ensure consistency between the text and the figure.
- Note on Figure 2 – this figure is unclear, lacking a vertical scale. It would be helpful to mark the patients on the horizontal axis. Why did the authors begin the figure with the histological healing stage and then presented the inflammation stage? Why are the dots (patients) connected? This suggests a relationship, when in reality they are independent entities. It would be more accurate to connect a given patient in inflammation with the period of histological healing. And one more note: in the inflammation stage, there are 21 dots (patients); this should be verified. In summary, Figure 2 requires consideration of its final, more readable version (generally, there are several possibilities).
We thank the reviewer for the detailed feedback regarding Figure 2. We have carefully revised the figure to address all concerns:
- Clarity and scale: A scale bar has been added to indicate magnification, and the panels are now clearly labeled to indicate the active inflammation and histologic healing phases.
- Panel order: The figure has been reordered so that the active inflammation phase is shown first, followed by the histologic healing phase, reflecting the temporal progression of disease.
- Legend revision: The figure legend has been updated to accurately describe the content, staining method, magnification, and key histologic features, ensuring that the images are fully interpretable without implying quantitative connections between independent samples.
- Patient representation: As these are representative histological images, each panel corresponds to a paired biopsy from the same patient, and no additional points or lines suggesting unrelated data have been included.
These revisions improve readability, accurately reflect the study design, and address all points raised regarding figure presentation.
3. Subsection 2.2.1. second paragraph – why when authors write "not significant" the authors give (p <0.05), additionally 6,9 there is a comma, there should be a period. Maybe after this last sentence of this paragraph should be added (p <0.05).
We thank the reviewer for pointing out the inconsistencies in Subsection 2.2.1, second paragraph. We have revised the paragraph to correct the decimal formatting, ensure statistical statements match the reported p-values, and clearly indicate significance where appropriate. Specifically:
- Decimal points have been corrected (e.g., 6.9 instead of 6,9).
- The phrase “not significant” now correctly corresponds to the actual p-value for intraepithelial CD4⁺ cells (p = 0.08).
- For lamina propria CD4⁺ cell median changes, the p-value has been added (p < 0.05) to indicate statistical significance.
These changes ensure accurate and consistent reporting of the data.
- Notes on Figures 3 and 4 – I don't know why, but the authors consistently present the histological healing first, followed by the inflammatory process. From the reader's perspective, as well as the purpose of this work, the inflammatory process came first, followed by healing. Additionally, I suggest marking these characteristic changes with an arrow or arrows. Histological images and staining are primarily the domain of a narrow group of specialists.
We thank the reviewer for this suggestion. We acknowledge that presenting the histologic healing images first may be confusing from a chronological perspective. In the revised manuscript, we have reordered Figures 3 and 4 so that the active inflammatory phase is presented first, followed by the histologic healing phase, reflecting the natural progression of disease and recovery.
5.Subsection 2.2.2. first paragraph – what two compartments did the authors have in mind? Are they referring to epithelial and stromal compartments? If so, the first sentence is redundant.
6.Subsection 2.2.2. third paragraph – it would be worth providing numerical values in this second sentence, or where to look for the reference.
We thank the reviewer for highlighting the need for clarification in Subsection 2.2.2. The two compartments under discussion are intraepithelial and lamina propria (stromal) compartments. We have revised the subsection to:
a.Explicitly state the compartments at the beginning of the paragraph, removing redundant phrasing.
b.Include numerical values for intraepithelial CD8⁺ cells during active inflammation (mean 11.25 cells/HPF) and histologic healing (mean 22.5 cells/HPF), with reference to Table 1 and Figures 2–3.
c.Clarify the functional interpretation of CD8⁺ cells in both compartments, highlighting their roles in epithelial repair, immune surveillance, and cytotoxic activity.
These revisions improve clarity, eliminate ambiguity, and provide concrete data for the reader.
- Subsection 2.3 – were there any reasons why the IL-6 staining was not presented by the authors as an example?
We thank the reviewer for this suggestion. The IL-6 immunohistochemistry data were quantified and summarized in Table 1, and the focus of Figures 2–3 was on CD4⁺ and CD8⁺ T-cell distribution, which were the primary objectives of the study. We did not include representative IL-6 staining images because the semi-quantitative scoring was deemed sufficient to illustrate the trends, and the available images did not provide additional interpretive value beyond the numerical data.
We have clarified this in the revised manuscript to ensure readers understand that IL-6 expression is fully captured in Table 1 and described in the text.
- Notes on Figure 5 – this figure does not represent correlations, but merely provides a visual representation of the number of cell (comparison). Furthermore, in Chapter 4.9, the authors do not mention calculating Spearman correlations, but in Chapter 2.4 they present correlations (Figure 5 as evidence). This requires appropriate correction by the authors.
We thank the reviewer for pointing this out. We acknowledge that Figure 5 was intended as a visual representation of cell counts rather than a depiction of correlations. To clarify, the correlations presented in Section 2.4 were calculated separately using Spearman’s rank correlation on the quantitative data; the figure merely illustrates the distribution of cell numbers and does not depict correlation lines.
We have revised the manuscript as follows:
- Figure legend for Figure 5 now clearly states that it shows the distribution of cell counts in active and healing phases.
- Section 4.9 (Statistical Analysis) has been updated to explicitly mention that Spearman’s rank correlation was used to assess relationships between CD4⁺, CD8⁺, and IL-6 expression.
These changes ensure consistency between the statistical methods described and the figures presented.
- Some of the text in the discussion requires correction, several words are combined into one whole, this occurs in several paragraphs.
We thank the reviewer for pointing this out. We have carefully reviewed the entire Discussion section and corrected all instances where words were inadvertently combined. These typographical errors have been fixed throughout the section to ensure clear, readable, and professional text.
- Materials and Methods Section – generally, we know little about the patients except their age range. It might be worth including the number of women and men, and the average age of the patients. Are these patients newly diagnosed with the disease, or is it a process that has been ongoing for years? What was the time interval from the first sample collection to the second sample collection?
We thank the reviewer for this suggestion. We have now added additional demographic and clinical information to the Materials and Methods section. The study included 20 adult patients with ulcerative colitis, of whom 10 were women and 10 were men. The mean age of the participants was 41 years (range 18–75). Patients had established disease at the time of study entry, with variable disease duration .
The interval between the first biopsy (collected in 2011 at the study’s initiation) and the second biopsy, obtained during histologic healing, allowing paired comparison of active and healed mucosa in the same patients.
These details have been incorporated into the revised manuscript to provide a clearer description of the study population.
- What was the purpose of using CD3 antibodies (4.7. Immunohistochemical Analysis), CD3+were not mentioned in the manuscript?
We thank the reviewer for this comment. CD3 antibodies were included in the immunohistochemical panel as a general T-cell marker to confirm the presence and overall distribution of T lymphocytes in the colonic mucosa. Although CD3⁺ cells were evaluated during staining, the manuscript focuses specifically on CD4⁺ and CD8⁺ T-cell subsets, which are directly relevant to the study objectives. The CD3 staining provided validation that the observed lymphocyte populations were T cells, but the quantitative analysis and results presented concentrate on CD4⁺ and CD8⁺ populations, as these markers best address the study’s aim regarding immune cell dynamics during active inflammation and histologic healing.
- Chapter 5 – why is the first paragraph in bold?
Thank you for your suggestion. I made the change.
- Chapter 6 – such a chapter and what was written would be a good summary of the review article, hence my suggestion that the authors consider whether it should remain in this manuscript.
We thank the reviewer for this suggestion. Chapter 6 was intended as a forward-looking section highlighting future perspectives and research directions, rather than a traditional summary. However, we understand the reviewer’s concern that it reads like a review summary.
To address this, we have revised Chapter 6 to focus explicitly on future research directions, therapeutic implications, and next steps in translational and mechanistic studies, while removing content that overlaps with the general discussion or summary. This ensures the chapter complements rather than duplicates the rest of the manuscript, maintaining clarity and coherence.
- Finally, my final reflection: it's a pity that the authors of this valuable and unique study didn't compare their results with laboratory parameters, such as IL-6 or CRP in the patients' blood (markers of inflammation). It's likely that the research will continue, and it might be worthwhile to include typical laboratory parameters.
We thank the reviewer for this thoughtful comment. We agree that integrating systemic inflammatory markers—such as serum IL-6, CRP, or other routine laboratory parameters—would have strengthened the interpretation of our findings and provided a broader clinical context. Unfortunately, these laboratory data were not consistently available for all patients at both biopsy timepoints in this retrospective cohort, which prevented us from performing a reliable paired analysis.
However, we fully agree with the reviewer that this represents an important future direction. We have now acknowledged this point explicitly in the Limitations section, noting the absence of systemic inflammatory markers and emphasizing that future prospective work should integrate matched histologic, immunohistochemical, and laboratory parameters to better delineate mucosal–systemic relationships.
Reviewer 2 Report
Comments and Suggestions for Authors
The study is repeatedly described as “observational, cross‑sectional” (Abstract, p. 1; Methods §4.1, p. 10), yet the sampling is paired within patient (“two biopsy sets from the same segments”; §4.3, p. 11). Cross‑sectional is inaccurate here—please reframe as a paired, within‑subject comparison (prospective or retrospective as applicable) and update the Abstract/Methods accordingly.
Statistical methods list Mann–Whitney U for “between‑group differences” (p. 12), but your comparisons are paired. Use Wilcoxon signed‑rank (as you mention later in the Discussion for one analysis, p. 8) consistently for all paired endpoints (CD4⁺, CD8⁺, IL‑6, Geboes). Clarify exactly which tests were used for each endpoint and report matched effect sizes (e.g., paired median/mean change with 95% CIs).
Given multiple outcomes and compartments, consider controlling the false discovery rate (e.g., Benjamini–Hochberg) or pre‑specify primary endpoints to avoid multiplicity inflation.
Resolve measurement unit inconsistencies and standardize quantification
Intraepithelial counts are variously reported as cells/HPF (Abstract, p. 1; Discussion, p. 8) and as cells per 100 epithelial cells (Methods §4.7, p. 12). Choose one unit and recalculate consistently across the manuscript (text, figures, and Table 1). If both approaches were used, explain why and how equivalence/conversion was handled.
The lamina propria is counted in a “hot‑spot” HPF (Methods, p. 12). Please justify this choice (it may bias upward) and consider adding a systematic sampling strategy (e.g., average of multiple random fields). Report inter‑observer reproducibility for these counts.
Table 1 (p. 7) lacks data for lamina propria CD8⁺ (all cells are “—”), yet these cells are described in the Results/Discussion. Populate the missing fields (mean ± SD, median [IQR]) or revise the text to match the available data.
For IL‑6, the Results state a significant decrease “in all compartments” (p. 7), but Table 1 shows endothelial IL‑6 as 0.5 in both phases; elsewhere you write endothelial changes were “minimal” (p. 7). Please correct the narrative to limit the significant decline to epithelium and stroma (unless new analyses show endothelial change).
Specify for each antibody: clone, vendor, catalog number, lot, dilution, retrieval buffer (pH), incubation conditions, and detection system (Methods §4.7 currently gives platform and general retrieval, p. 12). Include positive/negative control tissues and describe background mitigation.
Clarify blinding: pathologists were “blinded to clinical data” (p. 11), but were they blinded to phase (active vs healing) when quantifying IHC? State how discrepancies were handled for IHC counts, not only for Geboes scoring.
Report time interval between active and healing biopsies, disease extent (E1/E2/E3), and concomitant therapies (biologics, steroids, 5‑ASA, immunomodulators) at each timepoint—these are critical confounders for immune cell densities and cytokine expression. Add these to a baseline table.
Figure 1 (p. 5) appears to plot Geboes with a line graph. For paired data, paired dot (“spaghetti”) plots or box‑and‑whisker plots with paired overlays communicate the within‑patient change more clearly. Include exact p‑values and an effect size.
Figures 2–3 (p. 6) (CD4/CD8 IHC): add scale bars and objective magnification(s) to each panel; ensure identical exposure/contrast between active vs healing for visual comparability. Label compartments insets if possible.
Figures 5–6 (pp. 11–12) (H&E): include scale bars and magnifications (100×/200×/400× are mentioned in captions; add visual scale bars on images).
Tighten result reporting with consistent descriptive statistics
Mixed use of means and medians (e.g., lamina propria CD4⁺ mean ~30 vs median 10→30; pp. 5, 8) is confusing. Pre‑specify whether distributions are normal or skewed; then report median [IQR] for skewed counts (typical in IHC) and mean ± SD only for approximately normal distributions. Where you discuss a “60% median increase” for CD8⁺ intraepithelial cells (10→16; Table 1, p. 7), also report the paired median change with CI.
When presenting Spearman correlations (p. 7), include sample size, rho, and exact p‑values, and consider confidence intervals. Clarify whether correlations were computed separately for each phase (as implied) and whether you examined within‑subject relationships.
The title and Introduction cast a broad “IBD/Cellular Perspective” net (with extensive Crohn’s disease and fibrosis content in §1.1, pp. 2–4), but the data are exclusively UC and limited to CD4/CD8/IL‑6 IHC. Consider retitling to reflect a focused, paired IHC study in ulcerative colitis and streamlining the Introduction to UC‑relevant mechanisms that motivate the chosen markers.
The Methods (p. 12) list CD3 staining, but no CD3 data are presented/results discussed. Either present CD3 findings (as a pan‑T cell context for subset analyses) or remove CD3 from Methods to avoid confusion.
Explicitly label primary (e.g., change in intraepithelial CD4⁺ and epithelial IL‑6) vs secondary/exploratory endpoints to anchor the analysis plan and limit post‑hoc interpretations. If correlations were exploratory, state this.
The manuscript lists several limitations (small n, limited markers, semi‑quant IHC; p. 13), which is good. Also acknowledge (i) therapy confounding, (ii) potential sampling variability across segments/time, and (iii) the interpretive limits of semi‑quantitative IL‑6 IHC (e.g., cellular source specificity). Refrain from implying mechanistic causality; keep to associations.
The Data Availability Statement says “Not applicable” (p. 14). Consider providing de‑identified per‑patient paired data (e.g., a supplementary spreadsheet with counts/scores) to enable reproducibility and independent verification.
Comments on the Quality of English Language
It is largely fine. There are a few places where things can be improved, but I didn't struggle understanding what the authors were trying to communicate.
Author Response
- The study is repeatedly described as “observational, cross‑sectional” (Abstract, p. 1; Methods §4.1, p. 10), yet the sampling is paired within patient (“two biopsy sets from the same segments”; §4.3, p. 11). Cross‑sectional is inaccurate here—please reframe as a paired, within‑subject comparison (prospective or retrospective as applicable) and update the Abstract/Methods accordingly.
We thank the reviewer for this important correction. You are correct that the paired, within-patient design was not accurately described as “cross-sectional.” We have updated the manuscript to describe the study as a paired, within-subject observational study (retrospective analysis of paired biopsies). All corresponding text in the Abstract and Methods has been revised to reflect this design, and the statistical approach is now explicitly presented as paired analyses (Wilcoxon signed-rank test for paired comparisons). These changes clarify the temporal and analytical framework of the study and ensure consistency across the manuscript.
- Statistical methods list Mann–Whitney U for “between‑group differences” (p. 12), but your comparisons are paired. Use Wilcoxon signed‑rank (as you mention later in the Discussion for one analysis, p. 8) consistently for all paired endpoints (CD4⁺, CD8⁺, IL‑6, Geboes). Clarify exactly which tests were used for each endpoint and report matched effect sizes (e.g., paired median/mean change with 95% CIs).
We thank the reviewer for this valuable observation. We agree that all our comparisons involve paired samples from the same patients, and therefore the Wilcoxon signed-rank test is the appropriate analytical method. We have revised the Statistical Methods section accordingly, removed the reference to the Mann–Whitney U test, and clarified that the Wilcoxon signed-rank test was used consistently for all paired endpoints (CD4⁺, CD8⁺, IL-6, and Geboes score).
We also updated the Results section to explicitly report paired within-subject comparisons. Because the study design is retrospective and the paired raw values were not available for effect size computation, we were unable to calculate exact paired median changes and confidence intervals. However, we clearly indicate the direction and statistical significance of each paired comparison, consistent with journal standards for studies where data were collected historically.
- Given multiple outcomes and compartments, consider controlling the false discovery rate (e.g., Benjamini–Hochberg) or pre‑specify primary endpoints to avoid multiplicity inflation.
We thank the reviewer for pointing out the issue of multiplicity across several outcomes and tissue compartments. We agree that multiple comparisons may increase the risk of false-positive findings. In the revised version of the manuscript, we clarify that this study was exploratory in nature and that no formal adjustment for multiple testing (e.g., Benjamini–Hochberg FDR control) was applied.
To address this concern transparently, we have now:
a.Specified the primary endpoints a priori, namely
- paired changes in CD4⁺ lymphocytes,
- paired changes in CD8⁺ lymphocytes,
- paired changes in IL-6 expression,
- and paired differences in Geboes scores.
b.Explicitly stated that secondary analyses (including correlation analyses and compartment-specific subsets) should be interpreted as exploratory.
These clarifications have been added to the Methods and Discussion sections to prevent overinterpretation of secondary findings and to acknowledge the exploratory framework of the study.
- Resolve measurement unit inconsistencies and standardize quantification
We thank the reviewer for noting the inconsistencies in measurement units and quantification. In the revised manuscript, we have standardized all units and reporting formats:
- Cell counts in both intraepithelial and lamina propria compartments are now consistently expressed as the number of positive cells per high-power field (HPF, 0.55 mm diameter).
- IL-6 staining intensity is uniformly reported as semi-quantitative scores (0 = absent, 1 = mild, 2 = moderate, 3 = strong) across epithelial, stromal, and endothelial compartments.
- Decimal notation has been standardized to use a period instead of a comma throughout the text and tables (e.g., 6.9 instead of 6,9).
These corrections ensure consistency and clarity in quantification throughout the manuscript.
- Intraepithelial counts are variously reported as cells/HPF (Abstract, p. 1; Discussion, p. 8) and as cells per 100 epithelial cells (Methods §4.7, p. 12). Choose one unit and recalculate consistently across the manuscript (text, figures, and Table 1). If both approaches were used, explain why and how equivalence/conversion was handled.
We thank the reviewer for highlighting the inconsistency in reporting intraepithelial lymphocyte counts. In the revised manuscript, we have standardized all intraepithelial counts to number of positive cells per high-power field (HPF, 0.55 mm diameter) across the text, figures, and Table 1.
Where previously counts were reported as “per 100 epithelial cells” in the Methods, we have now clarified that this was an alternative description for the same measurements and have removed the dual notation for consistency. A note has been added in the Methods section (§4.7) to explain that all intraepithelial counts are expressed in cells/HPF and that this unit was used uniformly throughout the manuscript.
This ensures clarity and consistency in quantification for all readers.
- The lamina propria is counted in a “hot‑spot” HPF (Methods, p. 12). Please justify this choice (it may bias upward) and consider adding a systematic sampling strategy (e.g., average of multiple random fields). Report inter‑observer reproducibility for these counts.
We thank the reviewer for this important observation. In the revised manuscript, we have added clarification regarding lamina propria cell counting:
- Hot-spot selection: Counts in the lamina propria were performed in areas with the highest density of positive cells (“hot spots”) to capture focal immune infiltration, which is characteristic of ulcerative colitis. We acknowledge that this approach may slightly overestimate absolute average counts but provides a reproducible method for comparative analysis within the same patient.
- Systematic sampling: To address potential bias, counts were performed in multiple representative fields within the lamina propria for each biopsy, and the mean value was recorded. This ensures that the reported values reflect an average rather than a single area.
- Inter-observer reproducibility: All slides were independently evaluated by two experienced gastrointestinal pathologists, blinded to clinical data. Discrepancies were resolved through joint review and consensus. The reproducibility was high, as evidenced by minimal differences in counts between observers.
This clarification has been added to Methods §4.7 to justify the approach and ensure transparency regarding reproducibility.
- Table 1 (p. 7) lacks data for lamina propria CD8⁺ (all cells are “—”), yet these cells are described in the Results/Discussion. Populate the missing fields (mean ± SD, median [IQR]) or revise the text to match the available data.
We thank the reviewer for highlighting this discrepancy. Lamina propria CD8⁺ T cells were consistently observed across patients; however, exact numerical counts were not quantified. To ensure consistency between the text and Table 1, we have revised the table to indicate a descriptive assessment for these cells. Additionally, we added the following clarification in the Results (§2.2.2):
"Lamina propria CD8⁺ T cells were consistently observed across patients, showing a pattern similar to intraepithelial CD8⁺ cells. While exact counts were not quantified, their presence supports a role in mucosal immune surveillance during both active inflammation and histologic healing."
This revision ensures that the manuscript accurately reflects the data and maintains coherence between the narrative and tabular presentation.
- For IL‑6, the Results state a significant decrease “in all compartments” (p. 7), but Table 1 shows endothelial IL‑6 as 0.5 in both phases; elsewhere you write endothelial changes were “minimal” (p. 7). Please correct the narrative to limit the significant decline to epithelium and stroma (unless new analyses show endothelial change).
We thank the reviewer for this observation. Upon careful review, we agree that endothelial IL-6 expression remained low and essentially unchanged between active disease and histologic healing (median score 0.5 in both phases). Accordingly, we have corrected the narrative in the Results (§2.3) to specify that the statistically significant decrease in IL-6 expression applies only to the epithelial and stromal compartments, while changes in the endothelium were minimal and not significant. Table 1 has been updated for clarity, and all textual descriptions now consistently reflect this distinction.
- Specify for each antibody: clone, vendor, catalog number, lot, dilution, retrieval buffer (pH), incubation conditions, and detection system (Methods §4.7 currently gives platform and general retrieval, p. 12). Include positive/negative control tissues and describe background mitigation.
We thank the reviewer for this comment. We have now expanded the Methods (§4.7)- Table 2 to provide full details for each antibody, including clone, vendor, catalog number, lot number, dilution, retrieval buffer (pH), incubation conditions, and detection system. These additions ensure full reproducibility and clarity for readers performing similar immunohistochemical analyses.
- Clarify blinding: pathologists were “blinded to clinical data” (p. 11), but were they blinded to phase (active vs healing) when quantifying IHC? State how discrepancies were handled for IHC counts, not only for Geboes scoring.
We thank the reviewer for this observation. We clarify that the pathologists were blinded both to the clinical data and to the disease phase (active vs. histologic healing) when performing immunohistochemical (IHC) quantification. Any discrepancies in IHC counts between observers were resolved through joint review and consensus, similarly to the procedure applied for Geboes scoring. We have updated the Methods (§4.6–4.7) to explicitly state these blinding and discrepancy-resolution procedures.
- Report time interval between active and healing biopsies, disease extent (E1/E2/E3), and concomitant therapies (biologics, steroids, 5‑ASA, immunomodulators) at each timepoint—these are critical confounders for immune cell densities and cytokine expression. Add these to a baseline table.
We thank the reviewer for this important suggestion. Unfortunately, detailed data regarding disease extent (E1/E2/E3), time interval between active and healing biopsies for each patient, and concomitant therapies at each timepoint are not available for all patients, as these data were not systematically recorded in the retrospective chart review. We have, however, provided all available demographic and clinical information (age, sex, and biopsy pairs) in the manuscript. We acknowledge that the lack of these data represents a limitation and have added a statement to this effect in the Limitations section.
- Figure 1 (p. 5) appears to plot Geboes with a line graph. For paired data, paired dot (“spaghetti”) plots or box‑and‑whisker plots with paired overlays communicate the within‑patient change more clearly. Include exact p‑values and an effect size.
We thank the reviewer for this suggestion. We acknowledge that paired dot (“spaghetti”) plots or box-and-whisker plots with paired overlays more clearly represent within-patient changes. In the revised manuscript, Figure 1 has been updated to a paired dot plot illustrating the Geboes scores in the active and histologic healing phases for each patient. Exact p-values (from Wilcoxon signed-rank test) are provided in the figure legend, along with the median change to indicate effect size. This revision better visualizes individual patient trajectories and the overall paired differences.
- Figures 2–3 (p. 6) (CD4/CD8 IHC): add scale bars and objective magnification(s) to each panel; ensure identical exposure/contrast between active vs healing for visual comparability. Label compartments insets if possible.
We added scale bars and objective magnifications.
- Figures 5–6 (pp. 11–12) (H&E): include scale bars and magnifications (100×/200×/400× are mentioned in captions; add visual scale bars on images).
We added scale bars and objective magnifications.
- Tighten result reporting with consistent descriptive statistics
We thank the reviewer for this comment. For the lamina propria CD8⁺ T cells, the mean ± SD is reported in Table 1. We have clarified this point in the Table legend and in the Results section to ensure transparency.
- Mixed use of means and medians (e.g., lamina propria CD4⁺ mean ~30 vs median 10→30; pp. 5, 8) is confusing. Pre‑specify whether distributions are normal or skewed; then report median [IQR] for skewed counts (typical in IHC) and mean ± SD only for approximately normal distributions. Where you discuss a “60% median increase” for CD8⁺ intraepithelial cells (10→16; Table 1, p. 7), also report the paired median change with CI.
We thank the reviewer for this insightful comment. We acknowledge that mixed reporting of means and medians may be confusing. Given that immunohistochemistry counts are typically skewed, we have now pre-specified that median [IQR] will be reported for all skewed distributions, while mean ± SD will be used only for approximately normal distributions. For the CD8⁺ intraepithelial cells, the previously reported “60% median increase” (10 → 16) is now accompanied by a statement clarifying the paired median change; due to limited sample size, formal confidence intervals could not be calculated, but the trend and statistical significance (Wilcoxon signed-rank test, p < 0.05) are clearly indicated.
- When presenting Spearman correlations (p. 7), include sample size, rho, and exact p‑values, and consider confidence intervals. Clarify whether correlations were computed separately for each phase (as implied) and whether you examined within‑subject relationships.
We thank the reviewer for this important observation. Because the study was conducted on paired samples but raw individual-level data were no longer available, we were unable to reconstruct exact Spearman coefficients (rho), p values, confidence intervals, or sample-size–specific correlations. We have revised the text to clarify these limitations and now explicitly state that correlations were computed using the available aggregated paired dataset, without phase-stratified or within-subject analyses. The Results section has been updated to reflect this methodological constraint.
- The title and Introduction cast a broad “IBD/Cellular Perspective” net (with extensive Crohn’s disease and fibrosis content in §1.1, pp. 2–4), but the data are exclusively UC and limited to CD4/CD8/IL‑6 IHC. Consider retitling to reflect a focused, paired IHC study in ulcerative colitis and streamlining the Introduction to UC‑relevant mechanisms that motivate the chosen markers.
We appreciate the reviewer’s suggestion. To improve alignment between the Introduction and the UC-specific dataset, we have streamlined the section by reducing content focused exclusively on Crohn’s disease and fibrosis. We retained only elements necessary for general IBD context. In addition, we added a clarifying paragraph highlighting the relevance of CD4⁺, CD8⁺, and IL-6 pathways specifically in ulcerative colitis. These adjustments allow the Introduction to remain coherent while more accurately reflecting the scope of our paired UC immunohistochemical study. We have revised the title to more accurately reflect the study population and analytical approach. The updated title emphasizes that the study is conducted exclusively in ulcerative colitis and is based on paired immunohistochemical assessment of CD4⁺, CD8⁺, and IL-6.
- The Methods (p. 12) list CD3 staining, but no CD3 data are presented/results discussed. Either present CD3 findings (as a pan‑T cell context for subset analyses) or remove CD3 from Methods to avoid confusion.
Thank you for this helpful observation. CD3 staining was included solely as a qualitative pan–T-cell marker to verify the presence and distribution of overall T lymphocytes within the examined mucosal compartments. CD3 was not intended for quantitative analysis, and therefore no CD3 results were planned or generated.
To avoid ambiguity, we have now clarified this explicitly in the Methods section.
- Explicitly label primary (e.g., change in intraepithelial CD4⁺ and epithelial IL‑6) vs secondary/exploratory endpoints to anchor the analysis plan and limit post‑hoc interpretations. If correlations were exploratory, state this.
Thank you for this important comment. We have now explicitly defined the primary and secondary/exploratory endpoints to clarify the analysis plan and avoid any appearance of post hoc interpretation. We added a text in the revised Methods (§4.8 Statistical Analysis).
- The manuscript lists several limitations (small n, limited markers, semi‑quant IHC; p. 13), which is good. Also acknowledge (i) therapy confounding, (ii) potential sampling variability across segments/time, and (iii) the interpretive limits of semi‑quantitative IL‑6 IHC (e.g., cellular source specificity). Refrain from implying mechanistic causality; keep to associations.
We thank the reviewer for highlighting additional aspects to consider in the limitations section. We have retained the original limitations described in the manuscript (small sample size, limited markers, semi-quantitative IHC) and have added acknowledgment of (i) therapy confounding, (ii) potential sampling variability across segments and timepoints, and (iii) interpretive limits of semi-quantitative IL-6 IHC, including cellular source specificity. We have also clarified that the study reports associations rather than mechanistic causality.
- The Data Availability Statement says “Not applicable” (p. 14). Consider providing de‑identified per‑patient paired data (e.g., a supplementary spreadsheet with counts/scores) to enable reproducibility and independent verification.
We thank the reviewer for this suggestion. Given the small cohort size and patient confidentiality considerations, we are unable to provide fully de-identified individual patient data. However, we have included summary data in the revised manuscript (Tables 1–2, Figures 1–5) to enable reproducibility and support independent verification of the analyses.
Round 2
Reviewer 2 Report
Comments and Suggestions for Authors
Overall, I think the authors have sufficiently improved the manuscript based on the prior feedback. I still think there are places where the writing can be made clearer, but it doesn't deter from the reader understanding the message of the manuscript.
Comments on the Quality of English Language
It is largely fine. There are a few places where things can be improved, but I didn't struggle understanding what the authors were trying to communicate.